# Genome-Wide Identification and Expression Profiles of 13 Key Structural Gene Families Involved in the Biosynthesis of Rice Flavonoid Scaffolds

**DOI:** 10.3390/genes13030410

**Published:** 2022-02-24

**Authors:** Jianyong Wang, Chenhao Zhang, Yangsheng Li

**Affiliations:** State Key Laboratory of Hybrid Rice, Key Laboratory for Research and Utilization of Heterosis in Indica Rice, Ministry of Agriculture, College of Life Sciences, Wuhan University, Wuhan 430072, China; wangjianyong90@sina.cn (J.W.); zch_nx@126.com (C.Z.)

**Keywords:** flavonoid, *Oryza sativa*, gene family, phylogeny, evolution, stress

## Abstract

Flavonoids are a class of key polyphenolic secondary metabolites with broad functions in plants, including stress defense, growth, development and reproduction. *Oryza sativa* L. (rice) is a well-known model plant for monocots, with a wide range of flavonoids, but the key flavonoid biosynthesis-related genes and their molecular features in rice have not been comprehensively and systematically characterized. Here, we identified 85 key structural gene candidates associated with flavonoid biosynthesis in the rice genome. They belong to 13 families potentially encoding chalcone synthase (CHS), chalcone isomerase (CHI), flavanone 3-hydroxylase (F3H), flavonol synthase (FLS), leucoanthocyanidin dioxygenase (LDOX), anthocyanidin synthase (ANS), flavone synthase II (FNSII), flavanone 2-hydroxylase (F2H), flavonoid 3′-hydroxylase (F3′H), flavonoid 3′,5′-hydroxylase (F3′5′H), dihydroflavonol 4-reductase (DFR), anthocyanidin reductase (ANR) and leucoanthocyanidin reductase (LAR). Through structural features, motif analyses and phylogenetic relationships, these gene families were further grouped into five distinct lineages and were examined for conservation and divergence. Subsequently, 22 duplication events were identified out of a total of 85 genes, among which seven pairs were derived from segmental duplication events and 15 pairs were from tandem duplications, demonstrating that segmental and tandem duplication events play important roles in the expansion of key flavonoid biosynthesis-related genes in rice. Furthermore, these 85 genes showed spatial and temporal regulation in a tissue-specific manner and differentially responded to abiotic stress (including six hormones and cold and salt treatments). RNA-Seq, microarray analysis and qRT-PCR indicated that these genes might be involved in abiotic stress response, plant growth and development. Our results provide a valuable basis for further functional analysis of the genes involved in the flavonoid biosynthesis pathway in rice.

## 1. Introduction

Flavonoids, synthesized by the phenylpropanoid pathway, one of the largest families of polyphenolic secondary metabolites in the world, are extensively distributed in all different organs and tissues depending on the plant developmental and environmental conditions [1]. Nearly all flavonoids contain the common diphenylpropane (C6-C3-C6) carbon framework with two aromatic rings (A-ring and B-ring) interconnected by a three-carbon heterocyclic pyran ring (C-ring chain) [2,3]. Based on the degree of heterocyclic C-ring oxidation, the position of hydroxyl and methyl groups on the A- and B-rings and the degree of modifications (including glycosylation, acylation and polymerization ), flavonoids can be chiefly categorized into six classes: flavones, flavanones, flavonols, flavanols, anthocyanins and isoflavones [3,4]. An enormous amount of research during the last few decades has revealed that flavonoids, as the most bioactive plant secondary metabolites, might exhibit a wide range of physiological functions in plant growth, development, reproduction and stress defense [5,6,7]. As co-pigmentation, flavonoids in flowers and seeds can facilitate their pollination and seed dispersal through attracting insects [8,9]. As important developmental regulators, flavonoids have been shown to be involved in pollen fertility and pollen germination [10,11,12,13,14] and in phytohormone transport and catabolism regulation [11,15,16,17]. As antioxidants [18,19], flavonoids also play apparently important roles in abiotic and biotic stress responses, such as low phosphate and nitrogen stress [3,20], temperature stress [21,22,23], drought and salt stress [24,25,26], protection against ultraviolet-B (UV-B) radiation [27,28,29], as well as defense against pathogens and herbivores [30,31,32,33,34]. Furthermore, flavonoids also have demonstrated major benefits for human nutrition and health due to their diverse biological activities, such as anticancer, antiviral, anti-allergic, anti-inflammation and protection against coronary heart diseases [35,36,37]. Suffice it to say, it is increasingly important for crop yield security and human nutrition due to the antioxidant and healthy benefits of flavonoids to enhance flavonoid content in the future.

To date, over 9000 flavonoid molecules have been identified from various plant species [4], and the molecular mechanisms of enzymes catalyzing flavonoid biosynthesis have also been well established, such as in *Arabidopsis thaliana*, *Solanum lycopersicum* (tomato), *Glycine max*, *Phaseolus vulgaris* and *Vitis vinifera* [2,38,39]. *O. sativa* (rice), as an experimental system for monocots, is the most economically important food staple crop for human nutrition [35]. Although the understanding of flavonoid biosynthesis in rice is relatively limited when compared to dicot plants, such as Arabidopsis and tomato [2], a nearly complete flavonoid biosynthetic pathway for flavonoid scaffolds, including flavanones, flavones, flavonols and anthocyanins in rice is elucidated in Figure 1, based on several pioneering works [29,40,41,42,43,44,45,46,47]. The flavonoids are biosynthesized by a complex series of reactions, including condensation, isomerization, oxidations and reductions [2]. The first two processes in the flavonoid biosynthesis pathway are facilitated by chalcone synthase (CHS) and chalcone isomerase (CHI) to form the sequential production of chalcone [48] and flavanone (including naringenin) [49]. Flavanone is as a universal precursor for the biosynthesis of various flavonoid classes, including flavones, flavonols, anthocyanins and proanthocyanidins. Among them, flavones are synthesized by flavone synthase (FNS), which is divided into two types in plants, including FNS I (mainly in Apiaceae plants) and FNS II (more widespread in plants) [50]. In rice, flavanones are converted into corresponding flavones by OsFNS I-1 or OsFNS II in vitro or in vivo [42,51,52]. In addition, flavones can be also synthesized by flavanone 2-hydroxylase (F2H). F2H catalyzes the hydroxylation of flavanone to produces 2-hydroxyflavanones in the C-glycosylflavone biosynthesis pathway [40,46]. Dihydrokaempferol is synthesized by flavanone 3-hydroxylase (F3H) [53]. Then flavonol synthase (FLS) converts dihydroflavonols into flavonols by the desaturation of dihydroflavonol [54]. In rice, OsFLS is a bifunctional dioxygenase—exhibiting FLS and F3H activities to exercise its functions in flavonol production [44]. Based on the hydroxylation pattern of the flavonoid B-ring, flavonoid 3′-hydroxylase (F3′H) and flavonoid F3′5′H-hydroxylase (F3′5′H) catalyze hydroxylation at the C3′ and C3′ C5′ positions of flavonoid for the final production of two-hydroxy and three-hydroxy, respectively [55,56]. F3′5′H may represent the later gained functional divergence from F3′H [57,58]. To date, two rice F3′Hs, CYP75B3 and CYP75B4, were proven as bona fide flavonoid 3′-hydroxylase with substrate-specific specificity, both of which can restore Arabidopsis *transparent testa 7* mutants by catalyzing 3′-hydroxylation of flavonoids into 3′-hydroxylated flavonoids [41,43]. Notably, CYP75B4 (also termed chrysoeriol 5′-hydroxylase) can catalyze not only hydroxylation at the C3′ position of flavonoid but also the C5′ hydroxylation of chrysieriol, which is unique and indispensable for tricin formation in rice [41]. Finally, anthocyanin and proanthocyanidin biosynthesis begin with the entry step enzymatic activity of dihydroflavonol 4-reductase (DFR) [59,60]. DFR competing with FLS for the same substrate can catalyze the dihydroflavonols to produce corresponding leucoanthocyanidins, which are further converted into corresponding anthocyanidins with the aid of the enzymes leucoanthocyanidin dioxygenase/anthocyanidin synthase (LDOX/ANS) [61,62]. Leucoanthocyanidin reductase (LAR) and anthocyanidin reductase (ANR) convert leucoanthocyanidines and anthocyanidins, respectively, into their corresponding proanthocyanidins [63,64]. The flavonoid scaffold molecules mentioned above are further highly diversely modified, including through hydroxylation, C- or O-glycosylation, O-methoxylation and various other biochemical conversions with the aid of enzymes such as glycosyltransferases (GTs), acyltransferases and methyltransferases [38,57]. The carbon flux of the flavonoids pathways through a typical cell can represent about 20% of the total carbon flux in plants [2]. These ubiquitous flavonoid metabolic diversification processes in plants have been demonstrated to play essential roles in plants’ diverse environmental adaptation [57,58,65]. 

Gene duplication is a prevailing phenomenon in plant genomes and is considered as the major mechanism for forming gene families during evolution [67,68]. In some cases, increases in the total number of enzymes and production may be the result of the presence of multiple gene family members. In other cases, new functions can be formed by gene duplication. Both cases for the generation of new genetic material have been suggested to play critical roles in environmental adaptation of plants [69,70,71]. Therefore, genome-wide analysis of diverse gene families in plants can provide us with more new information and lay a foundation of important data for studies in future. Unlike Arabidopsis’ enzymes, which are encoded by single genes [38,72], several of the flavonoid-related biosynthesis enzymes in rice are encoded by multiple genes. The rice genome has been completely sequenced since 2005, and rice has been used universally as a species for functional genomics research due to its relatively small genome size (about 389 Mb) [73]. However, no systematic and comprehensive identification and analysis have been performed on the whole key structural gene families involved in the biosynthesis of the flavonoid scaffold molecules in the rice genome, except for studies on the CHS gene family [74,75]. Here, based on the recently released rice genome from the Rice Genome Annotation Project (RGAP) and the Rice Annotation Project (RAP) Network, genome-wide identification and functional analysis of the key structural enzymes involved in the biosynthesis of the flavonoid scaffold molecules are studied.

## 2. Materials and Methods

### 2.1. Identification, Sequence Feature and Phylogenetic Analysis of the Rice Key Flavonoid Gene Families

The rice reference genome database containing the protein and CDS sequences and GFF3 file were retrieved from EnsemblPlants (http://plants.ensembl.org/, accessed on 2 Septemper 2021). BLASTP searches were performed in the rice database by using the deduced protein homologues of the 13 flavonoid biosynthesis-related genes as a query sequence, including CHS, CHI, F3H, FLS, LDOX, ANS, FNS, F2H, F3′H, F3′5′H, DFR, ANR and LAR from various plant species downloaded from the National Center for Biotechnology Information (NCBI) database (Appendix A). A cut-off e-value of 1 × 10^−5^ was applied in the BLASTP method [76]. All redundant candidate protein sequences were removed, and all the nonredundant candidate proteins were verified by SMART (http://smart.embl-heidelberg.de/, accessed on 7 October 2021), CDD (https://www.ncbi.nlm.nih.gov/Structure/bwrpsb/bwrpsb.cgi, accessed on 7 October 2021) and Pfam (http://pfam.xfam.org/search/sequence, accessed on 7 October 2021). Next, the remaining confirmed protein sequences were further examined by HMMER searches (Appendix A) and blastp analysis against the NCBI and RAP databases with default parameters. Physicochemical parameters of all rice flavonoid-related candidate proteins, including molecular weights (MW) and theoretical isoelectric points (pI), were predicted by the ExPASy online tool server (https://web.expasy.org/protparam/, accessed on 20 November 2021) [77], and subcellular localizations were predicted via the WoLFPSORT server (https://wolfpsort.hgc.jp/, accessed on 23 November 2021). Multiple sequence alignments were conducted by using ClustalW with the default parameters [78]. An unrooted neighbor-joining (NJ) phylogenetic tree was generated by MEGA 7.0 with 1000 bootstrap replicates [79], and the final results were visualized using Evolview v2 (https://evolgenius.info//evolview-v2/, accessed on 30 November 2021) [80]. In addition, the corresponding second structure and sequence alignment from the flavonoid-related gene families were carried out by using the online servers PDB (https://www.rcsb.org/, accessed on 8 December 2021) [81] and ESPript (https://espript.ibcp.fr/ESPript/cgi-bin/ESPript.cgi, accessed on 8 December 2021) [82].

### 2.2. Gene Structure and Conserved Motifs

Gene structures (including exon/intron and domain) and their length data were extracted based on genome annotation GFF3 files and NCBI CDD, respectively. Conserved motifs were identified by using the online MEME program (http://meme-suite.org/tools/meme, accessed on 13 December 2021) with the following parameters: the maximum number of 20 motifs and motif width between 6 and 200 amino acids—other parameters were set at their default values [83]. Then, gene structure and conserved motifs were analyzed and visualized using TBtools [84].

### 2.3. Chromosome Location, Gene Duplication Events and Microsynteny Analysis

Chromosome location was obtained from genome annotation GFF3 files. Gene duplication events, including segmental and tandem duplication events, are the two principle means by which gene families are expanded [85]. Genes in the same subfamily were regarded as co-paralogs. Genes that were co-paralogs and located on duplicated chromosomal blocks within their genomes through polyploidy followed by chromosome rearrangement were defined as segmental duplicates. We defined tandem duplicates as adjacent co-paralogs within the same or neighboring intergenic region on a single chromosome. The gene duplication events were analyzed by using MCScanX software [86]. Circos software (http://circos.ca/, accessed on 3 December 2021) was used to visualize chromosome location and gene duplication events [87]. In order to analyze their selective force during the evolutionary process, the nonsynonymous (Ka) and the synonymous (Ks) substitution rates (that is Ka/Ks ratios) of duplicate gene pairs were calculated via DnaSP 5.0 (http://www.ub.edu/dnasp/, accessed on 3 December 2021) [88]. Divergence time of all duplicate events was estimated by T = Ks/(2 × 9.1) × 10^3^ million years ago (Mya) [89].

### 2.4. Plant Materials and Growth Conditions

Rice ‘RPY geng’ (*O. sativa* ssp. japonica) was used for quantitative real-time transcription polymerase chain reaction (qRT-PCR) analysis. Seedlings were germinated in water at 37 °C for 2 days then grown in containers with Yoshida solution under 14 h light (28 °C)/10 h dark (22 °C) with 65% relatively humidity. After two weeks, seedlings were transferred to Yoshida solution containing 200 mM NaCl or at 4 °C for analyzing salt and cold stress, respectively. Shoots were collected directly into liquid N_2_ and then stored at −80 °C prior to RNA isolation with three biological replicates (at least 15 seedlings/each replicate) at 0, 3 and 24 h for RNA extraction. Total RNA was extracted using TRIzol reagent (Invitrogen, Beijing, China), and first-strand cDNAs were synthesized by using an M-MLV reverse transcriptase reagent kit (Promega, Beijing, China) according to the manufacturer’s instructions. 

### 2.5. Expression Analysis of the Key Flavonoid Biosynthesis-Related Genes by Rice RNA-Seq and Microarray Datasets and qRT-PCR

Eleven transcriptome raw datasets from rice ‘Nipponbare’ (*O. sativa* ssp. *japonica*) were downloaded from NCBI Sequence Read Archive (SRA) databases (SRX100741, SRX100743, SRX100745, SRX100746, SRX100747, SRX100749, SRX100753, SRX100754, SRX100755, SRP151515, SRX020118 and SRX016110) to investigate the expression profiles of flavonoid biosynthesis-related genes in 11 different tissues, including leaves at 20 days, post-emergence inflorescence, pre-emergence inflorescence, anther, pistil, seed 5 days after pollination (DAP), embryos at 25 DAP, endosperm at 25 DAP, seed at 10 DAP, shoots at 14 days and seedlings at the four-leaf stage. Sequence reads were mapped to the reference genome (MSU 7.0, http://rice.plantbiology.msu.edu/, accessed on 16 December 2021) with Tophat2 [90]. Then, gene expression abundances were calculated with Cufflinks [91]. The gene expression patterns in the 11 various tissues were drawn via R/Bioconductor [92]. 

The microarray data of rice seedlings treated with cold (GSE57895) [93], salt (GSE76613) and various plant hormones (GSE39429) [94,95] were downloaded from the Gene Expression Omnibus (GEO, https://www.ncbi.nlm.nih.gov/geo/, accessed on 16 December 2021) database. R was used to perform the whole raw microarray data analysis to obtain the differentially expressed genes (DEGs) based on the conditions with a |log_2_fold change| > 1.5 and with significant results for the *t*-test (*p*-value < 0.05) after normalization to the control [96,97].

Rice salt- and cold-response genes (*OsCHS12*, *OsCHS28*, *OsF3H2*, *OsDFR6* and *OsLDOX2*) from rice RNA-Seq datasets were further examined through qRT-PCR. Primers of these five genes were designed by Primer 5.0 (Appendix A). The qRT-PCR reaction (10 μL) was formulated with three biological replicates in a 96-well plate by using the 2 × SYBR Green Master Mix reagent (Bio-Rad, Hercules, CA, USA). The rice actin was selected as an endogenous reference to normalize the expression levels of the target genes of all samples. The 2^−ΔΔCT^ method was used to calculate the relative amount of target gene expression.

## 3. Results

### 3.1. Identification and Characterization of Flavonoid Biosynthesis-Related Genes in Rice

The key genes encoding enzymes involved in flavonoid biosynthesis pathways were examined with a systematic computational approach through using BLASTP and the HMMER algorithm on the rice genome. As a result, a total of 85 genes were identified, belonging to 13 gene families (Table 1 and Appendix A), including 28 *CHS*, 10 *F3′H*, 9 *LDOX*, 8 *F3H*, 8 *DFR*, 7 *CHI*, 6 *ANR*, 3 *FLS* and 2 *ANS* genes, and *LAR*, *F3′5′H*, *FNSII* and *F2H* encoded by a single gene. All gene candidates were characterized and named according to their corresponding chromosomal order, and more detail information regarding their corresponding chromosome locations, number of amino acids (aa), molecular weight (MW) and theoretical isoelectric point (pI) are summarized in Table 1. Then, the subcellular localizations were also predicted, among which, the majority (43/85) were targeted to the cytoplasm, second (29/85) was the chloroplast and 4, 3, 2, 2 and 2 genes were located in the nucleus, extracellular, mitochondrion, plasma membrane and vacuole, respectively.

### 3.2. Analysis of Phylogenetic Relationship, Gene Structure and Motif Composition

In order to investigate evolutionary relationships, a NJ phylogenetic tree was constructed for the 85 putative genes in the rice flavonoid biosynthesis-related pathway, which revealed they were clustered into five main lineages (Figure 2). Among them, all *CHS* genes and *CHI* genes were separately clustered into single lineages, while the *F3H*, *FLS*, *LDOX* and *ANS* genes, *F3′H*, *F3′5′H*, *FNSII* and *F2H* genes, and *DFR*, *ANR* and *LAR* genes formed separate groups of lineages. Gene structure diversity has been proven to act as the primary driving force for multigene families’ evolution [98]. As expected, the gene structures and motif compositions were highly conserved in the same lineages, although they might show some differences in the number and the length of gene structure and motif composition in some cases. For instance, the Chal_sti_synt_N and Chal_sti_synt_C domains are conserved in all *CHS* genes. *CHI* genes possess a chalcone domain. *F3′H*, *F3′5′H*, *FNSII* and *F2H* genes contain the p450 domain. The adh_short or Epimerase or 3Beta_HSD domains are found in *DFR*, *ANR* and *LAR* genes, and the 2OG-FeII_Oxy or DIOX_N domain are contained in *F3H*, *FLS*, *LDOX* and *ANS* genes. These results suggest that 85 genes belonging to 13 gene families in the rice flavonoid biosynthesis pathway could be further divided into 5 five distinct lineages based on the analysis of gene structural features, motif compositions and phylogenetic relationships. Among them, F3H, FLS, LDOX and ANS are members of the 2-oxoglutarate dependent dioxygenase (2-ODD) superfamily, F3′H, F3′5′H, FNSII and F2H belong to the cytochrome P450-dependent monooxygenase (CYP450) superfamily, and DFR, ANR and LAR are subordinated to the short-chain dehydrogenase/reductase (SDR) superfamily (Figure 2).

### 3.3. Chromosome Locations, Duplication Events and Microsynteny Analysis

To understand the expansion mechanism, chromosome locations and duplication modes of the rice genes involved in the flavonoid biosynthesis were investigated by using MCScanX (Figure 3 and Table 2). We found that the majority of 85 flavonoid biosynthesis-related genes were distribute across Chrs 4, 6, 7, 9, 10 and 11. Among them, Chr4 and Chr10 both had 12 genes (14.12%), Chr7 owned 11 genes (12.94%), Chr9 included 9 genes (10.59%) and Chr6 contained 8 genes (9.41%). Remarkably, most of the genes, including not only those of the gene family members with high sequence similarity but also the different gene family members, were located close to each other on the Chrs. Some gene clusters with high sequence similarity were located in the same genome duplication zone as duplicate gene pairs and were found to be tandem duplicates (Figure 3 and Table 2), for instance *OsCHS4-7* on Chr5, *OsCHS11-12* and *OsLDOX24-26* in the region proximal to the telomere of Chr7 and Chr11, *OsANR1-5* on Chr4, *OsLDOX4-5* on Chr6 and *OsLDOX6-9* in the region proximal to the telomere of Chr9. In addition, gene clusters with a relatively high density among some distinct gene family members were also observed on Chrs 4, 6, 7, 8, 9 and 10. Based on previous reports [99,100,101], we assumed that a cluster of genes among different gene families may provide convenience in the form of physical position or space for distinct flavonoid-related genes to form macromolecular complexes during rice flavonoid biosynthesis. They included *OsF3H2-5* and *OsANR1-5* in the region proximal to the end site of Chr4, *OsF3H6* and *OsLDOX4-5* in the region proximal to the beginning site of Chr6, *OsDFR3* and *OsANS2/5* in the region proximal to the end site of Chr6, *OsCHI5* and *OsDFR4-5* in the region proximal to the end site of Chr7, *OsF3H7, OsDFR7* and *OsF3′H3/4* in the region proximal to the end site of Chr8, *OsDFR8* and *OsF3′H6-8* in the region proximal to the end site of Chr9, *OCHS16-21* and *OsF3′H9-10* in the region proximal to the beginning site of Chr10 and *OsF3H8* and *OsFLS2-3* in the region proximal to the end site of Chr10. Meanwhile, we also found that some of genes are located at the beginning or end site of Chrs, such as *OsF3H1*, *OsFNSII*, *OsCHS2*, *OsLODX3*, *OsF2H*, *OsCHI6* and *OsCHI7*, which are located at the beginning site of their corresponding Chrs, while *OsCHI3*, *OsF3H3*, *OsF3H4*, *OsF3H5*, *OsF3H8*, *OsFLS2* and *OsFLS3* mapped to the end site of their corresponding Chrs. 

For a gene family, gene duplication events, including tandem duplication, whole genome/segmental duplication, proximal duplication and dispersed duplication, are the primary power for gene expansion [85]. Therefore, 22 duplication events were examined in the total of 85 flavonoid-related genes, among which seven pairs were derived from segmental duplications and fifteen pairs were from tandem duplications (Figure 3 and Table 2). Furthermore, we noticed that these duplication events only existed in six gene families (*CHS*, *CHI*, *F3H*, *LDOX*, *ANS* and *ANR*), indicating that duplication events played an essential role in the expansion of these gene families. Further analysis displayed that the numbers and types of duplication modes were greatly diverse among these six gene families, which could mean that expansion mechanisms of these genes were different during the long-term evolutionary process. For example, tandem duplication was the principal duplication mode in the *LDOX* and *ANR* gene families, while segmental duplication was the major duplication mode in the *CHI*, *F3H* and *ANS* gene families. In addition, we found the number of tandem duplications was more than that of segmental duplications in the *CHS* gene family, which suggests tandem duplication played a principle role in the gene expansion.

Next, the values of non-synonymous substitutions per non-synonymous sites (Ka), synonymous sites (Ks) and the Ka/Ks ratio in each duplicated gene pair were calculated to evaluate the selective pressure. If a statistically significant Ka/Ks ratio equal to one represents a neutral or absence of selection, greater than or lower than that means positive and negative (or purifying) selection, respectively. The gene pair *OsLDOX4*/*OsLDOX5* was excluded to compute the Ka/Ks ratio because of values of Ka and Ks both equal to zero, which showed that OsLDOX4 and OsLDOX5 genes were completely identical at the nucleotide sequence level. Thereby, our findings showed the majority of Ka/Ks ratios were less than 1.0, meaning they underwent negative selection, except for five gene pairs, including *OsF3H1*/*OsF3H2, OsF3H2*/*OsF3H8*, *OsLDOX7*/*OsLDOX8*, *OsLDOX8*/*OsLDOX9* and *OsANS1*/*OsANS2*, which were greater than 1.0, suggesting positive selection (Table 2). At the same time, divergence time of all gene duplicate pairs varied from 0.55 to 27.64 Mya (Table 2). These results demonstrated that expansion of gene families in the flavonoid biosynthesis pathway was the result of multiple selective forces during the long-term evolutionary process.

### 3.4. Expression Profiling of Rice Genes in the Flavonoid Biosynthestic Pathway

Expression analysis of gene families can provide important research clues for the further study of their functional differentiation [102,103]. In this study, to investigate the expression pattern of the identified genes involved in the flavonoid biosynthesis, we obtained the expression profiles in 11 major tissues from the NCBI SRA databases, including leaves, post-emergence inflorescence, pre-emergence inflorescence, anther, pistil, seed at 5 DAP, embryos at 25 DAP, endosperm at 25 DAP, seed at 10 DAP, shoots and seedling at the four-leaf stage. We observed that the majority of the genes were expressed differentially in all tissues (Figure 4), except for 12 genes that were not detected in the all tissues. Most of the genes showed a relatively high expression level in specific tissues, while some genes had relatively high levels in multiple tissues. For instance, *OsFLS2*, *OsF3′H2*, *OsF3′H4*, *OsANR3*, *OsDFR3*, *OsDFR6* and *OsLDOX9* had relatively high expression in leaves. *OsCHS1*, *OsCHS8*, *OsCHS12*, *OsCHS18*, *OsCHS23*, *OsCHS26*, *OsCHS28*, *OsDFR1*, *OsF3′H3* and *OsLDOX8* showed high expression levels in seedlings at the four-leaf stage. *OsANS1*, *OsANS2, OsFLS1*, *OsF3H4*, *OsF3′H1* and *OsCHI4* were relatively highly abundant in anther. *OsF3H3* and *OsCHS9* had the highest relative expression levels in seed at 5 DAP. *OsCHS2*, *OsCHS24*, *OsCHS25*, *OsCHI3*, *OsANR5*, *OsDFR4*, *OsDFR5*, *OsDFR8*, *OsF3′5′H*, *OsF3’H9* and *OsFNSII* were relatively highly expressed in post-emergence inflorescence. *OsCHS10*, *OsCHS22*, *OsDFR7* and *OsANR1* were predominately expressed in pre-emergence inflorescence. *OsLDOX1*, *OsLAR1* and *OsDFR2* had relatively high levels in pistil. *OsLDOX5* and *OsF2H* had high expression levels in embryos at 25 DAP. *OsCHS13*, *OsCHS14*, *OsCHS15*, *OsLDOX2*, *OsLDOX7*, *OsF3H1*, *OsF3H2*, *OsF3H8*, *OsANR4*, *OsANR6* and *OsF3’H6* exhibited a high expression level in shoots. *OsCHS3*, *OsCHS27*, *OsCHI1*, *OsCHI2*, *OsCHI5*, *OsCHI6*, *OsCHI7*, *OsFLS3*, *OsFLS3*, *OsF3H5*, *OsF3H6*, *OsF3H7*, *OsLDOX3*, *OsLDOX4*, *OsF3′H8*, *OsF3′H10* and *OsANR2* were highly expressed in multiple tissues. In addition, all genes had relatively low expression levels in seeds at 10 DAP. These results indicate that gene families in the flavonoid biosynthesis pathway might be associated with distinct physiological functions in rice.

Many plant species are adversely affected by environmental stresses during their lifespan. Flavonoids, a group of the most bioactive plant secondary metabolites, play the most important roles in stress responses through serving as ROS scavengers by well-known antioxidant activities [5,22,23,39,65]. In the present study, gene expression profiles from the GEO database of rice seedlings under cold and salt stress were investigated to analyze the expression patterns of genes involved in the flavonoid biosynthesis pathway (Figure 5). Firstly, it should be noted that 29 genes were not detected in the two microarray datasets. Under cold stress, most of genes were upregulated in roots and shoots at almost all tested time points, which indicates flavonoids can play an important role in cold acclimation [22,23]. Unlike the cold stress, the majority of genes showed no response or were downregulated at almost all time points under salt stress, while some genes in roots and shoots were upregulated at some time points. For instance, *OsDFR6*, *OsANR6* and *OsLDOX2* were upregulated in roots and shoots at almost all time points. *OsCHS12* was upregulated in roots from 24 h salt stress. *OsLDOX1* and *OsF3H4* were upregulated for a mere 3 h in shoots after salt stress. *OsF3H7*, *OsLAR*, *OsCHS2* and *OsCHS27* were only upregulated in roots after salt stress for 24 h. *OsCHS28* and *OsF3H2* were upregulated in roots during the recovery. It is noteworthy that *OsCHS12*, *OsCHS28*, *OsF3H2*, *OsDFR6* and *OsLDOX2* were strongly upregulated under cold and salt stresses, suggesting that they may play critical roles in a plant’s adaption to cold- and salt-stress conditions. To further examine the accuracy of the above results, we analyzed these five key genes’ (*OsCHS12*, *OsCHS28*, *OsF3H2*, *OsDFR6* and *OsLDOX2*) relative expression levels via qRT-PCR under 200 mM NaCl and 4 °C treatment for 0 h, 3 h and 24 h (Figure 6). The qRT-PCR analysis results supported our conclusion mentioned above that the five genes (*OsCHS12*, *OsCHS28*, *OsF3H2*, *OsDFR6* and *OsLDOX2*) were cold- and salt-stress response genes.

It is commonly accepted that phytohormones play essential roles in the regulation of plant growth and development. The crosstalk between flavonoids and phytohormone pathways has been reported [11,15,16,17]. However, the function of the gene families from flavonoid-related pathways under phytohormone treatments has not been well-documented in rice. In order to explore it, the expression profiles from the NCBI GEO databases were obtained to analyze the genes’ relative expression patterns under 50 μM abscisic acid (ABA), 10 μM gibberellin (GA), 10 μM auxin (IAA), 1 μM brassinosteroid (BR), 1 μM cytokinin (CTK) and 100 μM jasmonic acid (JA) treatments (Figure 7). First, it is worth noting that 17 out of the total of 85 genes were not detected in the database. We observed that most of the genes were suppressed or downregulated in all six treatments, especially in GA, IAA and BR treatments, whereas some others were upregulated in one or multiple treatments (Figure 7). For instance, *OsF3H7*, *OsF2H* and *OsANR2* were found to be upregulated only under ABA treatment. *OsCHS10*, *OsCHS16*, *OsANS1*, *OsF3′H4* and *OsF3′H2* were only upregulated under CTK treatment. Expression of *OsCHS8*, *OsCHS13*, *OsCHS14*, *OsCHS27*, *OsCHI1*, *OsFLS3*, *OsANR3*, *OsDFR5*, *OsLDOX3*, *OsLDOX6* and *OsF3H2* were elevated solely under JA treatment. *OsLDOX2*, *OsCHI2*, *OsCHI4* and *OsCHS2* were upregulated when applied with ABA and JA treatments. *OsCHS6* and *OsCHS28* were upregulated following GA, CTK and JA exposure. Expression of *OsCHI6* and *OsCHI7* was higher under ABA and CTK. In addition, *OsCHS18* for CTK and JA, *OsF3H5* for IAA and CTK, *OsANS2* for ABA, GA and JA, *OsANR6* for ABA, CTK and JA, *OsCHS15* for BR, CTK and JA, *OsDFR3* for IAA and JA and *OsLDOX1* for ABA, IAA and JA were all upregulated. These findings demonstrate that these genes involved in flavonoid biosynthesis might participate in the relevant hormone signaling pathways.

### 3.5. Sequence Alignment and Phylogenetic Analysis of CHS Gene Family 

The CHSs belong to the representative type III polyketide synthase (PKS) superfamily, which is widely distributed in the plant kingdom from the lower bryophytes to angiosperms [57,104]. To date, at least one putative *CHS* gene candidate can be identified in every land plant species based on its corresponding available genomic data [57,105]. We found that the catalytic triad C164-H303-N336 inherited from the ancient PKS III superfamily [106] was highly conserved in almost all 28 *OsCHS* genes through using *Medicago sativa MsCHS* as a template to align the sequence of *OsCHSs* with the aid of the online servers PDB and ESPript (Appendix A). As the gatekeeper [106], phenylalanine at position 215 (F215) connected with CoA-binding was strictly conserved, while the other at F265 was relatively diverse. For instance, *OsCHS2*, *OsCHS11*, *OsCHS12*, *OsCHS24* and *OsCHS26* included a tyrosine (Y265), *OsCHS9*, *OsCHS13*, *OsCHS14* and *OsCHS15* contained glycine (G265), *OsCHS3* and *OsCHS28* included an isoleucine (I265), and *OsCHS25* contained a cysteine (C265) as a substitute for F265, which probably caused their functional diversity and divergent enzymatic activities of *OsCHS* genes. In addition, as a CHS/STS characteristic signature sequence [107], the WGVLFGFGP375GLT motif was also maintained in almost all 28 *OsCHS* genes (Appendix A), although their conserved motif mentioned above might show some substituted residues in some cases. Plant CHSs further fell into three distinct clades based on the phylogenetic relationships of OsCHSs and their homologs from other plant species on the NCBI (Appendix A). The majority of OsCHSs cluster into clade II, belonging to the oldest CHSs, except for six genes (*OsCHS1-3*, *OsCHS8* and *OsCHS27-28*) clustered into the largest clade I belonging to the typical CHS and 2 genes (*OsCHS10* and *OsCHS22*) clustered into clade III belonging to the members of anther-specific CHS-like (ASCL) proteins. Interestingly, all OsCHSs were more closely clustered with ZmCHSs, TaCHSs, HvCHSs and SiCHSs, indicating that the *CHS* homologs from Gramineae crops were more closely related.

### 3.6. Sequence Alignment and Phylogenetic Analysis of CHI Gene Family

CHIs are members of the CHI-folding protein family and can be categorized into four types based on sequence similarity, including types I–IV [108,109]. Generally, around 70% or more identical belong to the same type, while less than 50% identical are classified into different types. Type I CHI proteins are widespread in vascular plants, while type II CHI mainly exists in leguminous plants. Type III CHI proteins are the fatty-acid-binding proteins (FAPs) that participate in fatty-acid biosynthesis and are widely distributed in green algae and land plants [57]. Type IV CHI proteins are closely related to the ancestors of plant CHIs (CHI-like proteins (CHILs) that first appeared in mosses and evolved from FAPs) and are found in basal and higher plant species, including mosses, liverworts, lycophytes, ferns, gymnosperms and angiosperms [108,109]. In this study, we identified seven gene candidates encoding OsCHIs. Alignment of the OsCHI sequences to MsCHI showed that only OsCHI3 shared the highly conserved critical catalytic residues essential for CHI active sites, including R36, T48, Y106 and N113, except for serine at position 190 (S190) as a substitute for T190 (Appendix A), which was in line with what has been previously reported in rice. The mutation of OsCHI3 can cause the *golden hull* and *internode 1* phenotype [110]. However, the other six OsCHIs did not retain the catalytic core of the CHI protein fold and were substituted at many of these critical sites, which demonstrated they were divided into different types. A phylogenetic analysis of CHIs among plant species indicated that plant CHIs were grouped into four distinct clades (Appendix A), which was consistent with previous studies [108,109]. Among them, OsCHI3 belonged to type I, OsCHI6-7 clusterd to type IV, type II included OsCHI2, OsCHI4 and OsCHI5, and OsCHI1 was classified into type III, which was consistent with the results from alignment of the CHI sequences and analysis of their conserved residues. These findings indicated that the formed *OsCHI* gene family had undergone diverse functional divergence during the subsequent evolutionary process.

### 3.7. Sequence Alignment and Phylogenetic Analysis of 2OGD Gene Family (F3H, FLS, LDOX and ANS) 

Four flavonoid biosynthetic enzymes, F3H, FLS, LDOX and ANS, are members of the 2-oxoglutarate dependent dioxygenase (2-OGD) protein superfamily. The 2OGDs are localized in the cytosol and are non-heme, iron-containing proteins that can incorporate 2-oxoglutarate (2OG) and molecular oxygen (O_2_) into various substrates to form oxidized products (such as flavonoids), succinate and CO_2_ [111]. A phylogenetic analysis showed that plant 2-OGDs were divided into three classes, including DOXA, DOXB and DOXC [111]. F3H, FLS, LDOX and ANS were classified into the DOXC class, in which F3H was in the DOXC28, whereas FLS, LDOX and ANS were in DOXC47 [57,111]. In our study, 22 gene candidates, including eight *OsF3Hs*, three *OsFLS*, nine *OsLDOX* and two *OsANS* genes were identified (Table 1). Most of the rice *F3H*, *FLS*, *LDOX* and *ANS* genes were highly conserved in the 2-OGD specific and critical residue forming motifs (Appendix A), including the ferrous iron binding motif (H232-x-D234-xn-H288), 2-oxoglutarate binding motif (Y217 and R298-x-S300) and the proper folding motif of the 2-OGD polypeptide (G78, H85, P198, G272) [61,112]. In addition, the substrate binding motifs (F142, F144, K213, F304, E306) were only found in *OsFLS1*, which was consistent with recent findings [44]. Furthermore, the two FLS-specific motifs were also found in the three *OsFLSs*, except *OsFLS2* only included one motif due to the truncation of the C-end [113]. Interestingly, 12 residues relevant for F3H activity existed in almost all rice *F3H*, *FLS*, *LDOX* and *ANS* genes (Appendix A) [114], indicating that the *F3H* gene may be the ancestor of *FLS*, *LDOX* and *ANS* genes [57]. A phylogenetic analysis of the plants’ 2-OGDs aligned here grouped the genes into four distinct clades (Appendix A). Based on the phylogenetic analysis, all the remaining genes were included in the corresponding gene family, except *OsFLS2-3* clustered with *OsLDOXs* in clade IV, suggesting the *OsFLS* gene family performed a different function during the expansion process

### 3.8. Sequence Alignment and Phylogenetic Analysis of CYP450 Gene Family (F3′H, F3′5′H, FNSII and F2H)

F3′H, F3′5′H, FNSII and F2H are involved in flavonoid biosynthesis and belong to the huge cytochrome P450 (CYP450) superfamily of heme-containing membrane proteins localized on the cytosolic surface of the endoplasmic reticulum (ER) in eukaryotes [56]. CYP450s catalyze monooxygenase/hydroxylation reactions in various metabolic processes by insertion of an O atom using molecular O_2_ and NADPH as co-substrates [55,56]. CYP450s are generally categorized into families when having 40% or more amino acid sequence identity, and those are further grouped into subfamilies with 55% or more identity [57]. The F3′H and F3′5′H enzymes that are divided into the CYP75 family exist only in gymnosperms and angiosperms, while FNSII and F2H are members of the CYP93 family found solely in angiosperms [57,115]. In this study, ten *F3′Hs*, one *F3′5′H*, one *FNSII* and one *F2H* genes were identified from rice (Table 1). Sequence alignment showed that the majority of the rice *F3′H*, *F3′5′H*, *FNSII* and *F2H* genes contained the CYP450-featured conserved motif (Appendix A), including the proline-rich hinge region (LPPGPxxxP) required for the CYP450′s optimal orientation [116], the oxygen-binding pocket motif [117], the ExxR motif and PERF motif to form the E-R-R triad for core structure stabilization [118] and the heme-binding domain (FGxGRRxCxG) [119]. In addition, typical F3′H-specific motifs were also identified in all *OsF3′H* genes [58], such as the “VVVAAS” motif that has high similarity in both *OsF3′H9* and *OsF3′H10*, the “VDVKG” motifs that exist in almost all *OsF3′Hs* except for *OsF3′H6* (although there are amino acid substituted residues in some case). It should also be noted that six substrate recognition sites (SRSs) were also conserved in some or multiple sites, and the oxygen-binding pocket motif was included in SRS4 (Appendix A). The phylogenetic relationship of F3′H, F3′5′H, FNSII and F2H determined these proteins were divided into four clades. Among them, OsFNSII and OsF2H clustered with the typical FNSIIs and OsF3′5′H clustered with known F3′5′Hs, while OsF3′H1-8 and OsF3′H9-10 (CYP75B4 and CYP75B3) were separately divided into two distinct clades (clades I and II) (Appendix A) showing the functional divergence of OsF3′Hs during the evolutionary process. 

### 3.9. Sequence Alignment and Phylogenetic Analysis of SDR Gene Family (DFR, LAR and ANR)

DFR, LAR and ANR belong to the short-chain dehydrogenase/reductase (SDR) superfamily, which is the huge NAD(P)(H)-dependent oxidoreductase family ubiquitous in viruses, archaea, prokaryotes and eukaryotes [120]. Plant SDRs are categorized into five types: “classical”, “divergent”, “extended”, “atypical” and ”unknown”, based on their structures, motifs and active sites [121]. The DFR and ANR are member of the SDR108E family (“extended” type SDRs), while the LAR is classified into the SDR460A family in the “atypical” type [57,121]. We identified eight *OsDFRs*, six *OsANRs* and one *OsLAR* in rice via genome-wide analysis. The resulting alignment data indicated the catalytic core in these genes was highly conserved except for some variant residues (Appendix A). They included the glycine-rich Rossmann dinucleotide- (NADH/NADPH) binding motif, the putative conserved catalytic triad (S128, Y163, and K167) and the substrate binding region [59,60]. A phylogenetic analysis was performed by the homologs of DFR, LAR and ANR sequences (Appendix A) and revealed that these proteins were divided into five clades. Among them, OsLAR clustered with the known LARs, while the OsDFRs and OsANRs were separately divided into three and two distinct clades, respectively, indicating their functional divergence. OsDFR1 belonging to the typical DFRs was consistent with previous findings showing its role in the regulation of the rice pigmentation in hull and pericarp [122,123]. OsANR1-5 clustered with the characterized ANRs, while OsANR6 clustered with DFR-likes.

## 4. Discussion

It is well known that flavonoids play significant roles in a plant’s adaption to the diverse environmental conditions due to their potent antioxidant activity [5,39,65]. The metabolites and enzymes associated with the flavonoid biosynthesis pathway have been extensively investigated in dicotyledon plants, while only limited attention has been paid to monocots such as rice [2,58]. Given the fact that rice is a staple food, the identification and functional analysis of genes involved in flavonoid biosynthesis should be characterized comprehensively and systematically at the rice genome level. In the present study, we mainly focus on the key structural enzymes involved in the biosynthesis of the flavonoid scaffold molecules by using the recently released rice genome. 

In this study, 85 genes belonging to 13 gene families related to the rice flavonoid biosynthetic pathway were identified. On the basis of structural features, motif analyses and phylogenetic relationships, these 13 gene families were further subdivided into 5 distinct lineages, including the PKS III subfamily, CHI-folding protein family, 2-OGD superfamily, CYP450 superfamily and SDR superfamily (Figure 2). The majority of the 28 *OsCHSs* gene candidates were consistent with the previous studies with the exception of the *OsCHS19* (Os10g0168300) gene [74,75]. *OsCHS19* contained the typical Chal_sti_synt_N and Chal_sti_synt_C two domains and the characteristic catalytic triad (except for the C164) inherited from the PKS III ancestor, and the “gatekeeper” connected with CoA-binding at positions 215 and 256 were also highly conserved (Appendix A). Based on sequence features and conserved amino acid residue analysis, we proposed *OsCHS19* belonged to one of *CHS* gene families. Therefore, 28 *OsCHS* members were finally selected to analyze in this study. Notably, seven *OsCHS* genes, including *OsCHS4*, *OsCHS5*, *OsCHS7*, *OsCHS11*, *OsCHS17*, *OsCHS19* and *OsCHS20*, had no mRNA signal based on the expression profiles of RNA-Seq and microarray datasets, suggesting they might be pseudogenes (although these gene duplicates were retained during the evolutionary process). For *OsCHIs*, *OsCHI3* was in line with the recent finding of its belonging to the typical type I [110], while the remaining six *OsCHIs* were also examined and were classified as type II, III or IV (Appendix A). The finding indicated rice retained all four types of *CHI* during the subsequent evolutionary process. A recent study reported that two copies of *OsF3′H* (*CYP75B3* and *CYP75B4*) genes belonging to the CYP450 family presented in rice genome [41,43]. However, 10 *OsF3′H* gene copies were identified in this study, and *OsF3′H9* and *OsF3′H10* (also termed *CYP75B4* and *CYP75B3*) clustered with the typical *F3′Hs*, while the remaining eight *OsF3′Hs* clustered into the other clade (Appendix A), indicating the functional divergence of *OsF3′Hs* may be formed during the long-term evolutionary process via gene duplicates. Anthocyanin and proanthocyanidin are synthesized by DFR, LAR and ANR during the last steps of the pathway and are among the less understood flavonoid biosynthetic genes in rice. Although eight *OsDFRs*, six *OsANRs* and one *OsLAR* in rice were identified via genome-wide analysis in this study, their biological function remains unclear and needs to be further explored in the future. Recently, a widespread and systematic classification of the plant 2OGD superfamily has been performed [115]. Based on the study, *OsANS, OsF3H* and *OsFLS* genes in the rice genome were divided into the DOXC group, and it was proposed that only four rice *2-OGD* genes participated in the rice flavonoid biosynthesis pathway, including two *OsANSs* (Os01g0372500 and Os06g0626700), one *OsF3H* (Os04g0662600) and one *OsFLS* (Os02g0767300) [115]. Except for the *OsANS* genes, the others two genes were different from the results mentioned above, and eight *OsF3Hs* and three *OsFLSs* were identified in this study. Moreover, nine *OsLDOXs* were also examined. All these genes contained the 2-OGD specific and critical motifs and were categorized into their known corresponding gene families (except for OsFLS2-3, which clustered with the *OsLDOXs*) (Appendix A). Previous reports show naringenin can be converted to flavone in vitro by OsFNSIs (Os04g0581000 and Os03g0122300) [51,52]. It was difficult to distinguish the FNS I and F3H in rice due to their high amino acid homology. Therefore, the OsFNSIs were classified as *OsF3Hs* (termed OsF3H1 and OsF3H2) in our study based on phylogenetic analysis and the Rice Annotation Project Network. In addition, *OsF3H1* (Os03g0122300), mediating brown planthopper resistance via a flavonoid pathway, had also been confirmed in rice [30]. These findings suggest that more than four *2-OGD* genes are involved in the rice flavonoid pathway. We propose that these inconsistent results may be due to the limitations of previous identification methods by using older software or due to continuous updates to the rice genome over time. Notably, *OsFLS1* belonged to the conventional *FLS* gene family consistent with previous report in rice [44], while *OsFLS2-3* clustered with known *LDOXs* (Appendix A), indicating the *OsFLS* gene family had undergone functional divergence or is undergoing divergence through gene duplication, and the biological function of *OsFLS2-3* needs to be experimentally verified in the future. Furthermore, *OsFLS* and *LDOXs*/*ANSs* were closer in genetic relationship in contrast to *F3Hs*, and almost all of the rice *FLS*, *LDOX* and *ANS* genes existed the residues of *F3H* activity (Appendix A), suggesting that *F3H* may have been the first *2OGD* gene for flavonoid biosynthesis, and the other *2OGD* genes involved in flavonoid biosynthesis may have evolved from *F3H* [57]. 

It has been previously reported that flavonoids can be separately synthesized in cytoplasm (such as endoplasmic reticulum/ER), chloroplast and nucleus [18]. As expected, most of these 85 genes were predicted to be localized to the cytoplasm, followed by the chloroplast and nucleus in our study (Table 1). As it has been reported that exon–intron structures might have some regulatory functions in gene expression and evolution, absence of exons–introns might lead to fusion and rearrangement of disparate chromosome segments and difference of expression patterns in numerous gene families [98]. In this study, except for *OsF3′H3*, *OsF3′H5*, *OsF3′H7*, *OsANS1* and *OsANS2*, which had no intron, the other 80 genes separately contained 1, 2, 3, 4, 5 and 9 introns (Figure 2). These findings demonstrate that intron loss and gain may have occurred or was occurring during expansion of these gene families in rice. For instance, most of the 28 *OsCHS* genes contained a single intron, except for *OsCHS1*, *OsCHS5* and *OsCHS19*, which had two introns and *OsCHS11* had three introns (Figure 2), which was consistent with previous reports that most plant CHS family genes contain a single intron [124]. In seven gene candidates encoding the *OsCHIs* gene family, *OsCHI3* and *OsCHI5* contained three introns and *OsCHI1*, *OsCHI6* and *OsCHI7* had four introns, which was similar to other *CHI* genes, as they are generally composed of three or four introns (Figure 2). Notably, *OsCHI2* and *OsCHI4* had as many as nine introns, which is a rare phenomenon among gene structures but is similar to findings in cotton species [125]. Interestingly, we also found the distribution of 85 rice flavonoid-related genes on Chrs displayed a certain physical positon with bias to Chr4, Chr7, Chr9 and Chr10. Most of gene family members with high sequence similarity were clustered together on the Chrs in close proximity and were located in the genome duplication zone as duplicate gene pairs. These findings might be consistent with recent evidence that the rice genome has undergone genome duplication and expansion during the long-term evolutionary process, and tandem gene duplication represent the major proportion of the duplication events [126,127,128]. Several previous studies have shown that enzymes of the flavonoid biosynthetic pathway often assemble as complexes to facilitate metabolite channeling [99,100,101]. We found gene clusters with a relative high density were also observed among the distinct gene family members on Chrs 4, 6, 7, 8, 9 and 10, which might provide the chance for these genes to form macromolecular complexes in physical position or space during the rice flavonoid biosynthesis process. 

Earlier research demonstrated that gene duplication events, as a prevailing feature in plant genomes and include segmental and tandem duplication events, are the principle cause of the expansion of gene families [85]. With the aid of MCScanX software, we explored the expansion mechanisms of the 85 rice flavonoid-related genes and found 31 genes (31/85) were located in the rice genome duplication zone, including 7 segmental duplication pairs and 15 tandem duplication pairs (Figure 3 and Table 2). Notably, of these 85 tested genes, these duplication events existed only in six gene families (*CHS*, *CHI*, *F3H*, *LDOX*, *ANS* and *ANR*), while the remaining seven gene families (*FLS*, *DFR*, *LAR* and *CYP450s*) had no duplication events, which indicated that the expansion mechanisms of these 13 gene families were greatly different. We further noted that the numbers and types of duplication modes were greatly diverse among these six gene families which had duplication events. For instance, segmental duplications occurred in the expansion of the *CHI*, *F3H* and *ANS* gene families, while tandem duplication occurred in the *LDOX* and *ANR* gene families. Significantly, the instances of tandem duplication were more than that of segmental duplication in the *CHS* gene family, suggesting that tandem duplication was the dominant contributor to the expansion of the rice *CHS* gene family, which was consistent with the previous Han et al. (2017) investigation [75]. Meanwhile, we found the quantity of genes undergoing each duplication mode was directly correlated with that of the corresponding gene family. We speculated that expansion of these exclusive gene families may play essential roles for rice to adapt to the terrestrial environment during the process of evolution, and further verification was required to make the conclusion. Although no duplication events were identified in *DFR* and *F3′H* gene families via MCScanX, eight and 10 gene members, respectively, remained to be found separately in them, indicating that expansion of both gene families may be from other duplication modes, such as dispersed duplication, retroduplication, transposon-mediated duplication or proximal duplication. Taken together, these findings demonstrated that 13 flavonoid-related gene families in rice had undergone volatile, lineage-specific gene expansion. 

Previous studies reported that the origin time of the *Oryza* tribe (*Oryzeae*), the *Oryza* genus and the *Oryza*-AA genome were about 24, 15 and 5 Mya [129,130,131,132], respectively. In this study, the divergence times of all 22 gene pairs were estimated, and we found these duplication events occurred around between 0.55 and 27.65 Mya (Table 2), indicating the current flavonoid gene families in rice had at least four expansion events during the evolutionary process. Specifically, the divergence times of ten *OsCHS* gene pairs, including two segmental and eight tandem duplication events, were calculated to be a wider range of times, 7.19–27.65 Mya. Among them, two were between the origin of the *Oryza* genus and the diversification of *Oryzeae*, one was earlier than the origin of *Oryzeae* and the remaining seven were between the origin of the *Oryza*-AA genome and the diversification of the *Oryza* genus. These findings indicate that the *OsCHS* gene family in the rice genome was duplicated at least three times during the long-term evolutionary process, and the duplication events mainly occurred in the time of the *Oryza* genus differentiation. In addition, the divergence times of six duplication events from *OsF3H* and *OsANR* genes were estimated to be 16.04–24.26 Mya, indicating that these gene families’ duplications mainly occurred between the origin of the *Oryza* genus and the differentiation time of *Oryzeae*. Meanwhile, the estimated divergence time of three duplication events of *OsLDOXs* was 3.32, 6.36 and 16.51 Mya, close to the origin time of the *Oryza*-AA genome and the *Oryza* genus, suggesting that at least two duplication events of the *OsLDOX* gene families occurred during the recent differentiation of the *Oryza*-AA genome. Furthermore, the divergence time (0.55 Mya) of one *OsANS* duplication event was also calculated and found to be later than the origin of the *Oryza*-AA genome, indicating that expansion of OsANSs occurred recently in the differentiation of *Oryza*-AA genome. Based on these data, we summarized that gene duplications associated with the rice flavonoid synthetic pathway mainly occurred after the origin of the *Oryza* genus. Moreover, the ratio of Ka/Ks substitutions is used to determine the selective pressure after duplication. In our study, analysis of 22 duplicated gene pairs from the rice flavonoid biosynthetic pathway indicated that 16 (16/22) gene pairs were less than 1.0, while five gene pairs were greater than 1.0, which meant the majority of these genes underwent negative (or purifying) selection during the evolution (Table 2).

Biochemical characteristics, function and regulation can be reflected by distinct gene’s expression patterns in organisms [102,103]. The recently released high-throughput transcriptome data can provide valuable resources and information for analyzing rice functional genes at the genome-wide level. Given the close relationship between flavonoids and the plant’s environmental adaption, expression profiles of 85 genes related with flavonoid biosynthesis were detected in 11 various tissues and under several stresses (including phytohormone, cold and salt) (Figure 4, Figure 5, Figure 6 and Figure 7). In our study, most genes showed tissue-specific expression patterns (Figure 4), some of which were relatively high levels in multiple tissues (except for seed at 10 DAP), indicating that different gene families in the flavonoid biosynthesis pathway might have distinct physiological functions in rice. For instance, *OsANS1*, *OsANS2*, *OsFLS1*, *OsF3H4*, *OsF3′H1* and *OsCHI4* were relatively highly abundant in anther, while *OsFLS2*, *OsF3′H2*, *OsANR3*, *OsDFR3* and *OsLDOX9* had relatively high expression in leaves, which indicates these genes might play critical roles working together in anther and leaves. These results were similar to previous findings in other plant species [43,133,134]. In addition, plant species are adversely affected by abiotic or biotic stresses during their lifespan [135]. Previous studies revealed that genes associated with flavonoid biosynthesis participated in stress responses, such as cold and salt stresses [22,23,25,26]. Our result showed most genes strongly respond to cold stress (Figure 5), which is in line with the belief that flavonoids play important roles in cold acclimation [22,23,25]. Although the majority of genes were suppressed or had no response under salt stress (Figure 5), we still noted that some genes, such as *OsCHS12*, *OsCHS28*, *OsF3H2*, *OsF3H4*, *OsDFR6*, *OsANR6* and *OsLDOX1-2*, showed strong upregulation during salt stress and may act as good candidate genes for salt stress. Interestingly, *OsCHS12*, *OsCHS28*, *OsF3H2*, *OsDFR6* and *OsLDOX2*, with strong responses to cold and salt stresses, may play important roles in response to cold and salt stresses (Figure 5 and Figure 6), and their actual functions need to be further verified in the future. Further, genes involved in flavonoid pathways have been reported to be regulated by phytohormones [15]. However, the flavonoid-related genes’ response to phytohormone had not been studied in rice. In this study, we found that most genes were suppressed in six phytohormone treatments (Figure 7), especially under GA, IAA and BR treatments, which was consistent with the antagonized interaction between flavonoids and plant hormones [11,15,16,17]. Notably, there were some genes strongly responsive to plant hormone treatments, indicating these genes might participate in the relevant hormone signaling pathways. For instance, we found that *OsCHS10*, *OsCHS16*, *OsANS1*, *OsF3′H4* and *OsF3′H2* were upregulated under CTK treatment, while *OsF3H7*, *OsF2H* and *OsANR2* were upregulated under ABA treatment. Furthermore, expression level of *OsCHS8*, *OsCHS13*, *OsCHS14*, *OsCHS27*, *OsCHI1*, *OsFLS3*, *OsANR3*, *OsDFR5*, *OsLDOX3*, *OsLDOX6* and *OsF3H2* were elevated under JA treatment, which was in line with the previous findings from other plant species showing their role in the regulation of JA in flavonoid accumulation and the stimulation of flavonoid biosynthetic genes [133]. Taken together, the distinct expression patterns of genes associated with the flavonoid scaffold biosynthesis in this study provides some valuable information for further functional characterization. 

## 5. Conclusions

A systematic analysis of 85 genes associated with the rice flavonoid scaffold biosynthetic pathway, including chromosomal location, phylogenetic relationship, gene structure, motifs, duplication events, selective forces and expression patterns, was performed. The results revealed these 85 genes belong to 13 gene families and can be further categorized into five distinct lineages. Subsequently, 22 duplication events, including 7 segmental and 15 tandem, were identified, and it was found that flavonoid-related gene families in rice had mainly undergone lineage-specific gene expansion under purifying selection at different evolutionary time points, especially in the differentiation time of the *Oryza* genus. Moreover, expression patterns from RNA-Seq, microarray and qRT-PCR analysis demonstrated that genes involved in flavonoid scaffold biosynthesis showed spatial and temporal regulation in a tissue (or stress)-specific way, indicating they played important roles in plants’ growth, development and stress responses. We inferred *OsCHS12*, *OsCHS28*, *OsF3H2*, *OsDFR6* and *OsLDOX2* may be potential candidate genes for rice cold and salt resistance breeding, although their actual biological functions need to be explored in the future. 

## Figures and Tables

**Figure 1 genes-13-00410-f001:**
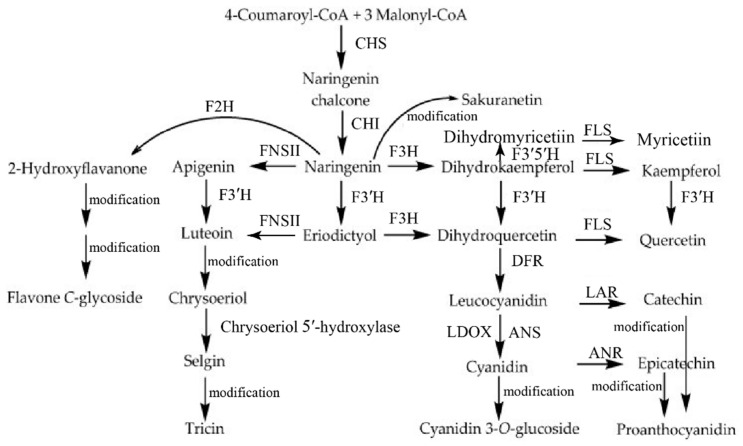
Proposed flavonoid biosynthetic pathway in rice [43,57,66]. Arrows represent enzymatic steps: CHS, chalcone synthase; CHI, chalcone isomerase; F3H, flavanone 3-hydroxylase; F2H, flavanone 2-hydroxylase; FNSII, flavone synthase II; F3′H, flavonoid 3′-hydroxylase; FLS, flavonol synthase; F3′5′H, flavonoid 3′5′-hydroxylase; DFR, dihydroflavonol 4-reductase; ANS, anthocyanidin synthase; LDOX, leucoanthocyanidin dioxygenase; LAR, leucoanthocyanidin reductase; ANR, anthocyanidin reductase; modification means the flavonoid scaffolds modification of hydroxylation, glycosylation, and methoxylation with the aid of enzymes including glycosyltransferases, acyltransferases and methyltransferases and so on.

**Figure 2 genes-13-00410-f002:**
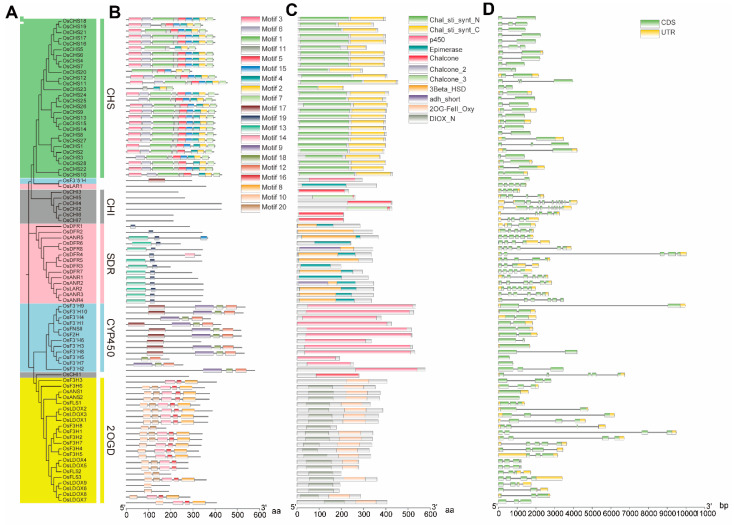
Phylogenetic tree (**A**), motif compositions (**B**), conserved domains (**C**) and exon/intron structure (**D**) of the 85 genes involved in the rice flavonoid scaffold biosynthetic pathway. (**A**) The neighbor-joining (NJ) phylogenetic tree was made by using MEGA 7.0 with 1000 bootstrap replicates. Different colors represent distinct lineages. (**B**–**D**) The widths of the boxes display relative lengths of genes and proteins. Differently colored boxes in (**B**,**C**) represent different motifs and domains, respectively. The green and yellow boxes and grey lines in (**D**) display exons, UTR and introns, respectively.

**Figure 3 genes-13-00410-f003:**
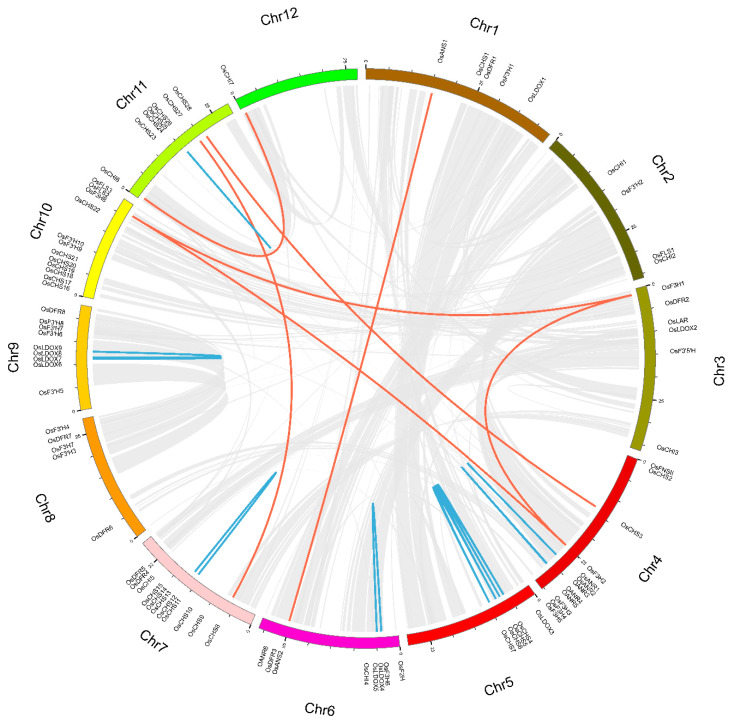
Chromosome location and duplication events of 85 genes associated with the rice flavonoid scaffold biosynthesis pathway in rice: 12 different chromosomes are displayed with 12 distinct colors. The segmental duplication and tandem duplication events are shown with red and blue lines, respectively, via Circos software.

**Figure 4 genes-13-00410-f004:**
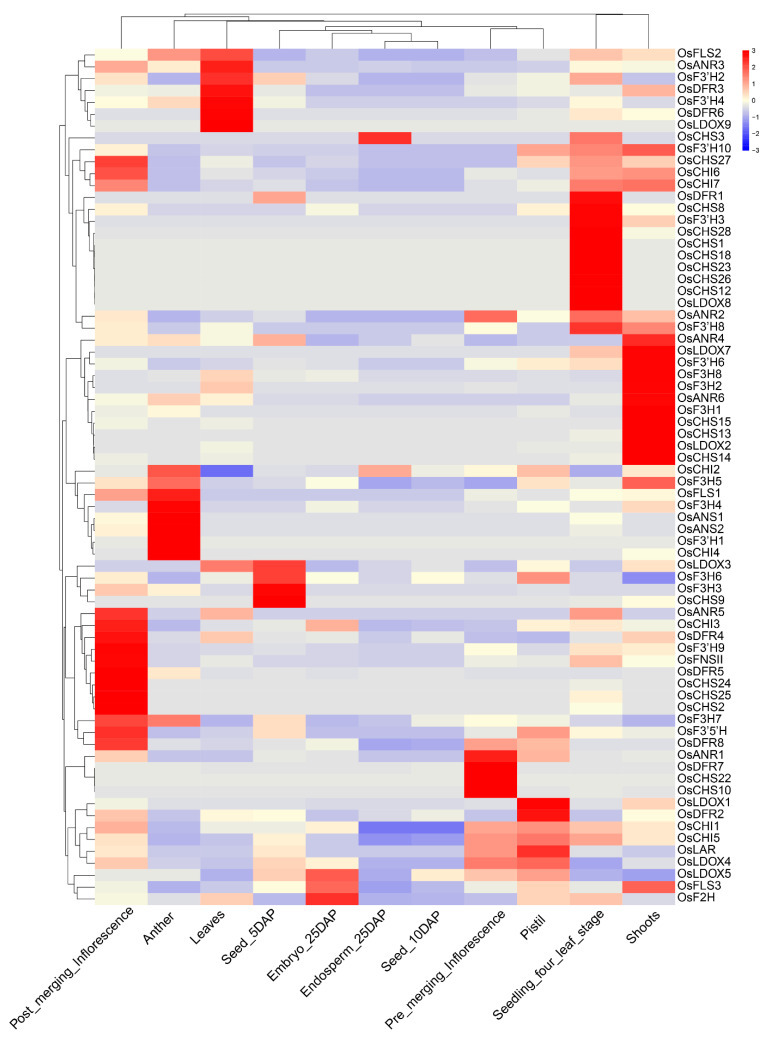
Clustering of expression profiles of 85 flavonoid-related genes in different tissues. The red/blue color indicates a relatively high/low expression level of transcript abundance, respectively: DAP, days after pollination.

**Figure 5 genes-13-00410-f005:**
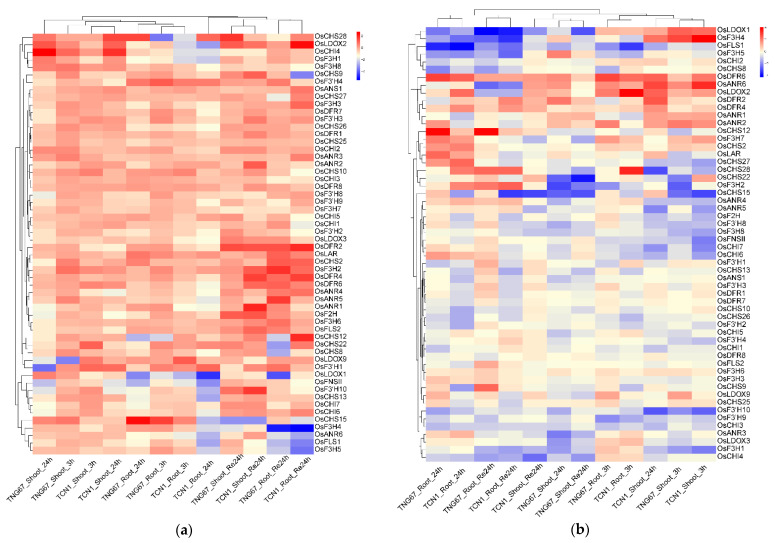
Expression profiles of 85 flavonoid-related genes under cold (**a**) and salt (**b**) stress: TNG67 and TCN1 represent two rice cultivars; 3 h, 24 h and Re24 indicates 3 h after treatment, 24 h after treatment and after a 24 h recovery, respectively. The values were calculated by log_2_foldchange based on the conditions, with significant results for the *t*-test (*p*-value < 0.05) after normalization to the control; red indicates upregulation and blue indicates downregulation.

**Figure 6 genes-13-00410-f006:**
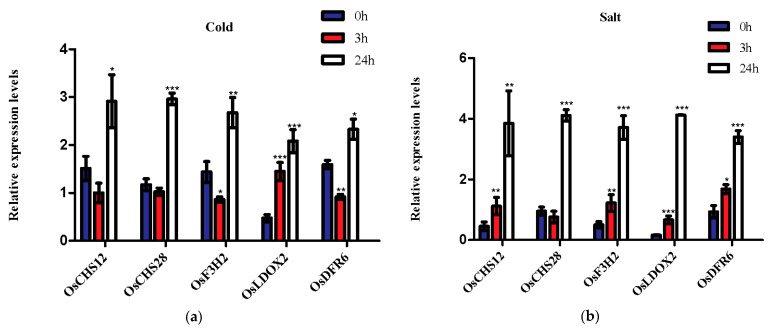
qRT-PCR analysis of the five cold- (**a**) and salt- (**b**) responsive genes involved in the biosynthesis of rice flavonoid scaffolds after 200 mM NaCl and cold (4 °C) treatment for 0 h, 3 h and 24 h. The relative expression values were calculated by 2^−ΔΔCT^ method. The bars indicate the standard deviation of 3 biological replicates; * shows a significant difference relative to the 0 h group (* *p* < 0.05, ** *p* < 0.01, *** *p* < 0.001).

**Figure 7 genes-13-00410-f007:**
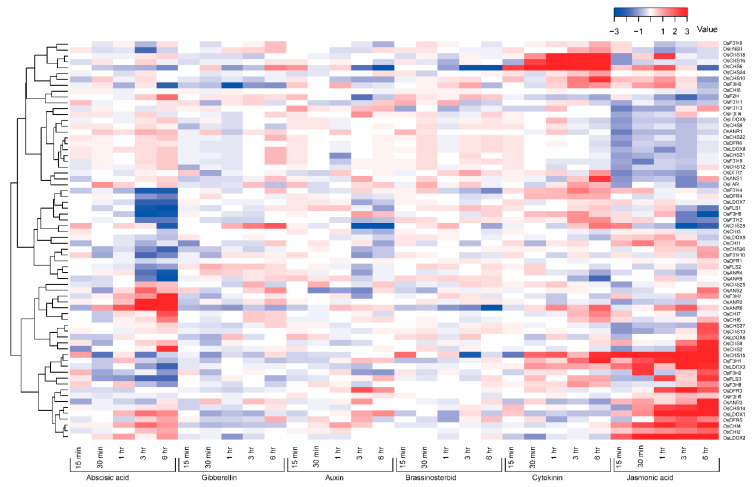
Expression patterns of 85 flavonoid-related genes under 6 hormone treatments: 15 min, 30 min, 1 h, 3 h and 6 h represent 15 min, 30 min, 1 h, 3 h and 6 h, respectively, after 50 μM abscisic acid (ABA), 10 μM gibberellin (GA), 10 μM auxin (IAA), 1 μM brassinosteroid (BR), 1 μM cytokinin (CTK) or 100 μM jasmonic acid (JA) hormone treatment. The values were calculated by the same method as those in Figure 5. Red and blue indicates upregulation and downregulation, respectively.

**Table 1 genes-13-00410-t001:** Characteristics of the 85 putative flavonoid biosynthesis genes in rice.

Lineage	GeneID	Gene Name	Chr^17^	Strat	End	Protein (aa)	MW^18^ (Da)	pI^19^	Predicted Location
CHS^1^	Os01g0602600	*OsCHS1*	Chr1	23688289	23692031	400	41801.7	6.71	Plasma Membrane
	Os04g0103900	*OsCHS2*	Chr4	258818	263006	394	42711	5.62	Extracellular
	Os04g0304600	*OsCHS3*	Chr4	13689982	13691362	374	39933.3	7.77	Cytoplasm
	Os05g0212500	*OsCHS4*	Chr5	6975843	6978049	393	42643.2	5.55	Cytoplasm
	Os05g0212600	*OsCHS5*	Chr5	6982114	6983529	313	33314.8	6.17	Cytoplasm
	Os05g0212900	*OsCHS6*	Chr5	6991336	6993702	393	42584.2	5.87	Cytoplasm
	Os05g0213100	*OsCHS7*	Chr5	7004068	7005454	393	42702.3	5.87	Cytoplasm
	Os07g0214900	*OsCHS8*	Chr7	6294613	6296305	404	43925	5.04	Cytoplasm
	Os07g0271500	*OsCHS9*	Chr7	10018726	10020736	403	43212	5.14	Cytoplasm
	Os07g0411300	*OsCHS10*	Chr7	12884544	12886091	430	46501.8	5.73	Cytoplasm
	Os07g0500800	*OsCHS11*	Chr7	18852472	18856432	455	49770	6.58	Chloroplast
	Os07g0501100	*OsCHS12*	Chr7	18873790	18875926	406	42833.5	6.25	Cytoplasm
	Os07g0525500	*OsCHS13*	Chr7	20420138	20421516	399	42972.6	6.11	Chloroplast
	Os07g0525900	*OsCHS14*	Chr7	20455512	20456848	399	42593.1	6.19	Cytoplasm
	Os07g0526400	*OsCHS15*	Chr7	20504681	20506402	400	42386.8	5.71	Cytoplasm
	Os10g0158400	*OsCHS16*	Chr10	3696797	3698774	399	43206.8	6.27	Cytoplasm
	Os10g0162856	*OsCHS17*	Chr10	4226902	4229142	399	43285.9	6.02	Cytoplasm
	Os10g0167900	*OsCHS18*	Chr10	4680918	4682894	400	43218	6.51	Chloroplast
	Os10g0168300	*OsCHS19*	Chr10	4700276	4701808	344	37489.3	6.09	Cytoplasm
	Os10g0168500	*OsCHS20*	Chr10	4725371	4726288	295	31618.9	4.79	Chloroplast
	Os10g0177300	*OsCHS21*	Chr10	5334799	5336232	364	39213.5	6.18	Cytoplasm
	Os10g0484800	*OsCHS22*	Chr10	18316325	18318781	390	42228.1	5.36	Cytoplasm
	Os11g0529100	*OsCHS23*	Chr11	19207697	19208437	212	22183.4	7.15	Mitochondrion
	Os11g0529500	*OsCHS24*	Chr11	19226735	19228499	414	43890.6	6.06	Plasma Membrane
	Os11g0529800	*OsCHS25*	Chr11	19236424	19238358	402	42389.1	6.32	Extracellular
	Os11g0529900	*OsCHS26*	Chr11	19243252	19244840	408	42592.2	6.83	Mitochondrion
	Os11g0530600	*OsCHS27*	Chr11	19277557	19281016	399	43264.3	6.19	Cytoplasm
	Os11g0566800	*OsCHS28*	Chr11	21102794	21104586	400	42942.8	6.59	Cytoplasm
CHI^2^	Os02g0320300	*OsCHI1*	Chr2	12765282	12772010	280	29656.8	9.06	Chloroplast
	Os02g0778500	*OsCHI2*	Chr2	32957456	32961345	427	47783.2	7.52	Cytoplasm
	Os03g0819600	*OsCHI3*	Chr3	34394508	34395638	233	23892.8	4.94	Cytoplasm
	Os06g0203600	*OsCHI4*	Chr6	5227416	5231610	428	47577.5	8.6	Nucleus
	Os07g0571600	*OsCHI5*	Chr7	23076409	23078827	263	28521.5	10.02	Chloroplast
	Os11g0116300	*OsCHI6*	Chr11	729514	732755	211	23397.5	4.49	Cytoplasm
	Os12g0115700	*OsCHI7*	Chr12	766441	768578	211	23487.6	4.55	Cytoplasm
2OGD^3^	Os03g0122300	*OsF3H* ^6^ *1*	Chr3	1235460	1244932	342	38738.7	6.9	Nucleus
	Os04g0581000	*OsF3H2*	Chr4	29331563	29338260	340	38823.8	5.44	Cytoplasm
	Os04g0662600	*OsF3H3*	Chr4	33809616	33812412	406	44413.2	10.64	Chloroplast
	Os04g0667200	*OsF3H4*	Chr4	34060040	34063474	326	36840.6	6.27	Cytoplasm
	Os04g0667400	*OsF3H5*	Chr4	34067654	34070800	333	37187	6.39	Chloroplast
	Os06g0162500	*OsF3H6*	Chr6	3145477	3147598	352	38634.3	5.88	Chloroplast
	Os08g0480200	*OsF3H7*	Chr8	23723615	23727247	337	36999	5.61	Cytoplasm
	Os10g0536400	*OsF3H8*	Chr10	20886407	20892109	178	20176.8	7.2	Cytoplasm
	Os02g0767300	*OsFLS* ^7^ *1*	Chr2	32310799	32312184	331	36818.7	6.46	Cytoplasm
	Os10g0559200	*OsFLS2*	Chr10	22012573	22014287	200	22850.7	5.06	Nucleus
	Os10g0559500	*OsFLS3*	Chr10	22019715	22023118	360	40369.9	4.83	Cytoplasm
	Os01g0832600	*OsLDOX* ^8^ *1*	Chr1	35636303	35640933	366	40477.4	5.06	Cytoplasm
	Os03g0289800	*OsLDOX2*	Chr3	10039387	10044161	387	41500.5	6.93	Cytoplasm
	Os05g0127500	*OsLDOX3*	Chr5	1561984	1568161	368	40298.2	6.32	Cytoplasm
	Os06g0176500	*OsLDOX4*	Chr6	3851134	3852336	278	31767.1	4.92	Cytoplasm
	Os06g0177700	*OsLDOX5*	Chr6	3889190	3890392	278	31767.1	4.92	Cytoplasm
	Os09g0353400	*OsLDOX6*	Chr9	11288774	11291404	190	21304.7	4.35	Cytoplasm
	Os09g0353700	*OsLDOX7*	Chr9	11308997	11310729	406	46245.9	11.15	Chloroplast
	Os09g0354100	*OsLDOX8*	Chr9	11320344	11323083	287	32216.6	6.74	Cytoplasm
	Os09g0354300	*OsLDOX9*	Chr9	11348309	11350044	194	21671.5	4.39	Cytoplasm
	Os01g0372500	*OsANS* ^9^ *1*	Chr1	15346352	15347941	375	40707.6	5.4	Cytoplasm
	Os06g0626700	*OsANS2*	Chr6	25296195	25297316	373	40711.6	5.99	Cytoplasm
CYP450^4^	Os01g0700500	*OsF3’H* ^10^ *1*	Chr1	28991472	28993283	426	47475.7	8	Cytoplasm
	Os02g0503100	*OsF3’H2*	Chr2	17805404	17808857	578	62941.7	11.02	Chloroplast
	Os08g0456200	*OsF3’H3*	Chr8	22369333	22371011	520	58459.3	8.43	Chloroplast
	Os08g0547900	*OsF3’H4*	Chr8	27479062	27481059	379	41614.5	6.02	Cytoplasm
	Os09g0263933	*OsF3’H5*	Chr9	4737258	4737839	193	21399.7	5.69	Cytoplasm
	Os09g0441600	*OsF3’H6*	Chr9	16392870	16394251	335	38630.5	9.81	Chloroplast
	Os09g0441625	*OsF3’H7*	Chr9	16395700	16396467	255	28607.8	5.82	Chloroplast
	Os09g0441700	*OsF3’H8*	Chr9	16403807	16407995	531	60585.5	8.63	Chloroplast
	Os10g0317900	*OsF3’H9*	Chr10	8494248	8504238	535	58518	7.18	Chloroplast
	Os10g0320100	*OsF3’H10*	Chr10	8679311	8681266	526	57516.8	8.15	Chloroplast
	Os03g0367101	*OsF3’5’H* ^11^	Chr3	14381298	14382935	294	31374.2	9.97	Vacuole
	Os04g0101400	*OsFNSII* ^12^	Chr4	99797	101630	516	57109.8	7.71	Chloroplast
	Os06g0102100	*OsF2H* ^13^	Chr6	163259	165320	518	56680	8.29	Chloroplast
SDR^5^	Os01g0633500	*OsDFR* ^14^ *1*	Chr1	25382714	25384678	284	31218.2	4.86	Chloroplast
	Os03g0184550	*OsDFR2*	Chr3	4441960	4443828	341	36876	7	Chloroplast
	Os06g0651000	*OsDFR3*	Chr6	26650225	26652370	198	21189	6.76	Chloroplast
	Os07g0601100	*OsDFR4*	Chr7	24517974	24527995	335	36852	5.67	Chloroplast
	Os07g0602000	*OsDFR5*	Chr7	24567545	24570285	340	37429.8	6.5	Chloroplast
	Os08g0183900	*OsDFR6*	Chr8	4897613	4900341	243	26555.6	8.91	Nucleus
	Os08g0515900	*OsDFR7*	Chr8	25591265	25593041	295	32462.8	6.62	Chloroplast
	Os09g0491788	*OsDFR8*	Chr9	18994543	18998420	343	37901.3	5.3	Vacuole
	Os04g0630100	*OsANR* ^15^ *1*	Chr4	32049063	32051694	321	34321.5	10.11	Chloroplast
	Os04g0630300	*OsANR2*	Chr4	32059560	32062390	346	36960.7	4.81	Chloroplast
	Os04g0630400	*OsANR3*	Chr4	32063756	32065731	346	37580.5	4.76	Extracellular
	Os04g0630800	*OsANR4*	Chr4	32085612	32088272	344	37539.4	4.83	Chloroplast
	Os04g0631000	*OsANR5*	Chr4	32131925	32135388	336	36369.2	4.82	Chloroplast
	Os06g0683100	*OsANR6*	Chr6	28470174	28472020	367	39555.6	7.5	Cytoplasm
	Os03g0259400	*OsLAR* ^16^	Chr3	8406809	8408296	358	38895.9	5.3	Cytoplasm

CHS^1^, chalcone synthase; CHI^2^, chalcone isomerase; 2OGD^3^, 2-oxoglutarate dependent dioxygenase; CYP450^4^, cytochrome P450; SDR^5^, short-chain dehydrogenase/reductase; F3H^6^, flavanone 3-hydroxylase; FLS^7^, flavonol synthase; LDOX^8^, leucoanthocyanidin dioxygenase; ANS^9^, anthocyanidin synthase; F3′H^10^, flavonoid 3′-hydroxylase; F3′5′H^11^, flavonoid 3′5′-hydroxylase; FNSII^12^, flavone synthase II; F2H^13^, flavanone 2-hydroxylase; DFR^14^, dihydroflavonol 4-reductase; ANR^15^, anthocyanidin reductase; LAR^16^, leucoanthocyanidin reductase; Chr^17^, chromosome location; MW^18^, molecular weight; pI^19^, theoretical isoelectric point.

**Table 2 genes-13-00410-t002:** Ka/Ks and divergence time analysis for the duplicated paralogs in rice flavonoid pathway.

Seq_1	Seq_2	Ka	Ks	Ka/Ks	Duplication Type	Date (Mya^3^)	Purifying Selection
*OsCHS3*	*OsCHS28*	0.053	0.324	0.163	SD^1^	17.81	YES
*OsCHS4*	*OsCHS5*	0.035	0.209	0.168	TD^2^	11.46	YES
*OsCHS4*	*OsCHS7*	0.009	0.187	0.049	TD	10.29	YES
*OsCHS5*	*OsCHS6*	0.039	0.167	0.233	TD	9.17	YES
*OsCHS6*	*OsCHS4*	0.022	0.174	0.126	TD	9.56	YES
*OsCHS6*	*OsCHS7*	0.022	0.244	0.090	TD	13.38	YES
*OsCHS8*	*OsCHS27*	0.087	0.503	0.173	SD	27.64	YES
*OsCHS11*	*OsCHS12*	0.062	0.241	0.258	TD	13.24	YES
*OsCHS24*	*OsCHS25*	0.090	0.131	0.688	TD	7.19	YES
*OsCHS25*	*OsCHS26*	0.248	0.393	0.631	TD	21.60	YES
*OsCHI6*	*OsCHI7*	0.019	0.156	0.122	SD	8.57	YES
*OsF3H1*	*OsF3H2*	0.402	0.293	1.376	SD	16.07	NO
*OsF3H1*	*OsF3H8*	0.198	0.326	0.608	SD	17.90	YES
*OsF3H2*	*OsF3H8*	0.379	0.349	1.086	SD	19.15	NO
*OsLDOX4*	*OsLDOX5*	0.000	0.000		TD		
*OsLDOX6*	*OsLDOX9*	0.253	0.301	0.843	TD	16.51	YES
*OsLDOX7*	*OsLDOX8*	0.072	0.061	1.195	TD	3.33	NO
*OsLDOX8*	*OsLDOX9*	0.171	0.116	1.478	TD	6.36	NO
*OsANR1*	*OsANR2*	0.118	0.292	0.403	TD	16.04	YES
*OsANR2*	*OsANR3*	0.214	0.442	0.485	TD	24.26	YES
*OsANR4*	*OsANR5*	0.125	0.327	0.381	TD	17.97	YES
*OsANS1*	*OsANS2*	0.024	0.010	2.327	SD	0.55	NO

SD^1^, mean segmental duplication; TD^2^, mean tandem duplication; Mya^3^, mean million years ago.

## Data Availability

Not applicable.

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
