# Peer review of "Genome-Wide Identification and Expression Profiles of 13 Key Structural Gene Families Involved in the Biosynthesis of Rice Flavonoid Scaffolds"

_genes, 2022, doi:10.3390/genes13030410_

Round 1

Reviewer 1 Report

This manuscript presents an interesting study on genome-wide identifies the functional genes in rice flavonoid biosynthesis. In general, the manuscript is well written and the conclusions are consistent with the data shown. Nevertheless, the manuscript presents also some shortcomings that should be corrected.

  1. In the introduction, line 127, the authors indicated that the rice genome size is about 364 Mb, but the actual size should be 389 Mb. It is recommended the authors to revise it and provide the reference.
  2. In table 1, the authors should be provided complete information, such as the enzyme abbreviations used in the table should be explained in the footnote.
  3. The scale bars in figure 2B, C, D, should be provided units.
  4. In the figure 2D, I am curious about that some genes are close to 9-10kb in size and there are transposable element-related sequences in the gene structure?
  5. It is recommended that the author increase the size of the text in the figure, such as figure 4, 5, and 6, to increase the readability for the reader.
  6. Some typing errors need to revise, such as line 424, tyoe, figure s7, recognizatioin.

Author Response

Response to Reviewer 1 Comments

We are pleased with the reviewers’ positive notes as well as critical but constructive comments to strengthen our manuscript. Our responses are marked by red font. New additions and changes in the main text are also marked by red font.

Point 1: In the introduction, line 127, the authors indicated that the rice genome size is about 364 Mb, but the actual size should be 389 Mb. It is recommended the authors to revise it and provide the reference.

Response 1: We has been corrected to “389 Mb” in the revised manuscript and we cited one original publication (Kawahara et al. Rice (New York, N.Y.) 2013, 6 (1), 4.) according to the Reviewer’s comments.

Point 2: In table 1, the authors should be provided complete information, such as the enzyme abbreviations used in the table should be explained in the footnote.

Response 2: We have provided the explaintions for the abbreviations used in table 1 and they were added into the footnote of table 1 according to the Reviewer’s suggestion.

Point 3: The scale bars in figure 2B, C, D, should be provided units.

Response 3: We are very sorry for our negligence of units of the scale bars in figure 2B, C, D, and We have respectively provide units “aa” for figure 2B and C and “bp” for figure 2D in the figure 2 of revised manuscript.

Point 4: In the figure 2D, I am curious about that some genes are close to 9-10kb in size and there are transposable element-related sequences in the gene structure?

Response 4: In this study, we found three genes including OsDFR4, OsF3’H9 and OsF3H1 are over 9kb in size based on the Reviewer’s comments. It is really true as Reviewer suggested that a large number of repeat regions were predicted in these genes structure via the RepBase Update server (https://www.girinst.org/repbase/update/), such as Helitrons, LTRs, Mariners, indicating that these genes own transposable element-related sequences in gene structure which probably caused their functional diversity and divergent enzymatic activities,and their biological function need to be further explored in the future.

Point 5: It is recommended that the author increase the size of the text in the figure, such as figure 4, 5, and 6, to increase the readability for the reader.

Response 5: Considering the Reviewer’s suggestion, we have increased a higher font size and clarity of the figure 4, 5, and 6 for the readability.

Point 6: Some typing errors need to revise, such as line 424, tyoe, figure s7, recognizatioin.

Response 6: We are very sorry for our incorrect writing, and we have very comprehensively and carefully corrected the spelling errors in the revised manuscript.

Reviewer 2 Report

In the present study, the authors identified 85 key structural genes in rice flavonoid biosynthesis, used a diverse bioinformatics analysis methods. The MS supported information for flavonoid biosynthesis in rice. 

My major concern is as follows:
For the first time identified the flavonoid gene families, you are used Blast function, 13 flavonoid biosynthesis-related genes as a query sequence. But in rice, only these 13 genes could represent all flavonoid genes? There no 14? 15?  

And the 13 genes are including all features or characteristics of the flavonoid gene? Because I think this study is a little different from other similar type studies of gene family research, such as WRKY family, GATA family, if using WRKY box or GATA domain as the query sequence or using Pfam model, this is a reasonable method, because these all gene family members contained consistent feature (such as WRKY or GATA domain), so Blast or HMM search could include all family members. Return to this study, the 85 genes don’t have consistent domain or feature, only using 13 reported genes is not enough for genome-wide research, please consider this. And also, Subject is functional Gene or Gene family? 

1. In title, there showed “genes in rice flavonoid biosynthesis”, which means these genes already contained function about flavonoid biosynthesis? I think it does not.
2. In abstract, “we identified 85 key structural genes…. They are classified into 13 genes….”, if combined with method 1st part, because you used 13 functional genes (already reported) to joined blast processing, so it showed 13 groups, this is obvious results.
3. “Our results provide valuable messages for comprehensive understanding the flavonoid biosynthesis pathway in rice.” I think this sentence is unsuitable because the function is not clear.

Author Response

Response to Reviewer 2 Comments

We express our sincere gratitude to the reviewers for their critical but constructive comments attempts to strengthen the manuscript. Our responses (red fonts) follow the reviewers’ comments.

Point 1: My major concern is as follows:For the first time identified the flavonoid gene families, you are used Blast function, 13 flavonoid biosynthesis-related genes as a query sequence. But in rice, only these 13 genes could represent all flavonoid genes? There no 14? 15?  And the 13 genes are including all features or characteristics of the flavonoid gene? Because I think this study is a little different from other similar type studies of gene family research, such as WRKY family, GATA family, if using WRKY box or GATA domain as the query sequence or using Pfam model, this is a reasonable method, because these all gene family members contained consistent feature (such as WRKY or GATA domain), so Blast or HMM search could include all family members. Return to this study, the 85 genes don’t have consistent domain or feature, only using 13 reported genes is not enough for genome-wide research, please consider this. And also, Subject is functional Gene or Gene family? 

Response 1: To our best knowledge, the molecular mechanisms of enzymes catalyzing flavonoid scaffolds biosynthesis including flavanones, flavones, flavonols, and anthocyanins have been well established in rice (Figure 1). If we consider the modification of the flavonoid scaffold molecules including hydroxylation, C or O-glycosylation, O-methoxylation and various other biochemical conversions with the aid of enzymes such as glycosyltransferases (GTs), acyltransferases, and methyltransferases, more genes (more than 13 genes) would be involved in the rice flavonoid biosynthetic pathway indicating a greater attempts and larger workload will be added. Because 13 flavonoid biosynthesis-related genes in the study can represent the general flavonoid scaffolds biosynthetic pathway in rice based on previous several pioneering works, here, we finally focus on the 13 gene families involved in the rice flavonoid scaffolds melecules biosynthesis in this study. In addition, our study mainly focus on the gene family involved in the flavonoid scaffolds biosynthetic pathway, of course, this metabolic pathway may be involve in multiple gene families which is different from other type studies of gene family most of which are just focus on a single gene family. Besides, except CHS and CHI gene families possess exclusive corresponding Pfam models, others have no corresponding specific Pfam models but share the same domain or Pfam model, such as, F3′H, F3′5′H, FNSII and F2H genes all contain the p450 domain, and the adh_short, Epimerase or 3Beta_HSD domains are found in the DFR, ANR and LAR genes, and the 2OG-FeII_Oxy or DIOX_N domain are contained in the F3H, FLS, LDOX and ANS genes in our study (Figure 2). Considering the Reviewer’ comments, 13 gene families were further examined by using HMM search based on the results mentioned above, which is in line with our previous results, indicating our results could include all 13 gene family members by blast homology searches. Moreover, 13 reported genes cann’t represent all flavonoid genes but can represent the general flavonoid scaffolds biosynthetic pathway in rice, so we have finally corrected the title of the manuscript and “functional gene”of the previous title is replaced by “Gene family” according to the Reviewer’s suggestion.

Point 2: In title, there showed “genes in rice flavonoid biosynthesis”, which means these genes already contained function about flavonoid biosynthesis? I think it does not.

Response 2: We have made correction the previous title of “Genome-Wide Identification and Functional characterization of Key Structural Genes in Rice Flavonoid Biosynthesis ” to “Genome-Wide Identification and Characterization of Key Structural Gene Families Involved in the Biosynthesis of Rice Flavonoid Scaffolds” according to the Reviewer’s comments.

Point 3:  In abstract, “we identified 85 key structural genes…. They are classified into 13 genes….”, if combined with method 1st part, because you used 13 functional genes (already reported) to joined blast processing, so it showed 13 groups, this is obvious results.

Response 3: To date, 13 flavonoid biosynthesis-related genes could represent the genes of general flavonoid scaffolds biosynthetic pathway in rice. Therefore, 13 gene families were finally selected to analyze in this study. As Reviewer suggested that 13 groups is obvious results, so the sentence of “we identified 85 key structural genes…. They are classified into 13 genes….” in abstract has been changed to “we identified 85 key structural gene ... They belong to 13 families…”.

Point 4: “Our results provide valuable messages for comprehensive understanding the flavonoid biosynthesis pathway in rice.” I think this sentence is unsuitable because the function is not clear.

Response 4: Considering the Reviewer’s suggestion, this sentence of “Our results provide valuable messages for comprehensive understanding the flavonoid biosynthesis pathway in rice.” in abstract has been corrected to “Our results provide valuable messages for further functional analysis of these genes involved in the flavonoid biosynthesis pathway in rice.”

Round 2

Reviewer 2 Report

Thank you for the revision, I respected the detailed supplement and explanation. But still, I have a different idea for this study.

I think this study contained two major points.

The first, Identification, though the authors already explained the function or relationship of the 13 reported genes. But I still worried about the “widely”, 85 genes is enough or not enough widely for the trait.
Because as shown in the author's response, this study is widely, or not specific one gene family, please upload the HMM results in Supp. Table, I think the results could support some evidence for this concern.

Second, the function characterization. In this study, all of the analysis is based on bio-informatics, there is no experiment, even the expression pattern of some key gene, qRTPCR should be performed for Provement of data results. Only the database is not enough to get some conclusion. 
Also there only expression data results showed the evidence for function conclusion (Involved in the Biosynthesis of Rice Flavonoid Scaffolds), most of the other results showed gene structure, family member relationship, etc… 
I feel this is inappropriate.

Also, some minor concerns:

1.    Line 174. Duplication, how to defined? Should add something in the text.
2.    In my downloaded PDF version, there have two Fig.2.
3.    In my downloaded PDF version, Fig.4, Fig.5, and Fig.6 look like strange, 2 legends?
4.    Line 389, please add the detailed information of each phytohormone, such as concentration, name, etc...

In short, the first point is still the major concern, because it is the origin of the subject, up to now, I think this study is not enough for publish in Genes.

Author Response

We express our sincere gratitude again to the reviewers for the continued critical but constructive comments attempts to improve the manuscript. Our responses are marked by red font. New additions and changes in the main text are also marked by red font using the “Track
Changes” function in MS Word.

Point 1: Thank you for the revision, I respected the detailed supplement and explanation. But still, I have a different idea for this study.I think this study contained two major points.The first, Identification, though the authors already explained the function or relationship of the 13 reported genes. But I still worried about the “widely”, 85 genes is enough or not enough widely for the trait. Because as shown in the author's response, this study is widely, or not specific one gene family, please upload the HMM results in Supp. Table, I think the results could support some evidence for this concern. Second, the function characterization. In this study, all of the analysis is based on bio-informatics, there is no experiment, even the expression pattern of some key gene, qRTPCR should be performed for Provement of data results. Only the database is not enough to get some conclusion. Also there only expression data results showed the evidence for function conclusion (Involved in the Biosynthesis of Rice Flavonoid Scaffolds), most of the other results showed gene structure, family member relationship, etc… I feel this is inappropriate. In short, the first point is still the major concern, because it is the origin of the subject, up to now, I think this study is not enough for publish in Genes.

Response 1: This comment is very valuable and helpful for revising and improving our paper. Considering the Reviewer’s suggestion, our final results mainly concern on the 13 gene families involved in the rice flavonoid scaffolds melecules biosynthetic pathway which were grouped into five distinct lineages including CHS (PF00195 and PF02797), CHI (PF02431, PF16035 and PF16036), 2OGD (PF14226 and PF03171), CYP450 (PF00067) and SDR (PF00106, PF01370 and PF01073) gene families. Among them, F3′H, F3′5′H, FNSII and F2H genes all contain the p450 domain belonging to CYP450 gene family, and the adh_short (PF00106), Epimerase (PF01370) or 3Beta_HSD (PF01073) domains are found in the DFR, ANR and LAR genes belonging to SDR gene family, and the 2OG-FeII_Oxy (PF03171) or DIOX_N ((PF14226) domain are contained in the F3H, FLS, LDOX and ANS genes belonging to 2OGD gene family in our study (Figure 2). Pfam models downloaded from the Pfam database were then used to query the predicted corresponding gene families by using HMM search according to the Reviewer’s comments. 28 OsCHSs, 7 OsCHIs, 95 OsOGDs, 334 OsCYP450s and 199 OsSDRs gene families were identified via HMM search. Notably, except for CHS and CHI gene families, 11 others have no corresponding specific Pfam models, these 11 predicted corresponding protein sequence obtained from 2OGD (PF14226 and PF03171), CYP450 (PF00067) and SDR (PF00106, PF01370 and PF01073) Pfam models searches need to be further examined by blastp analysis against the NCBI and RAP database with default parameters to finally comfirm the corresponding subfamilies (Supp. Table S6). Furthermore, 5 key genes including OsCHS12, OsCHS28, OsF3H2, OsDFR6 and OsLDOX2 reponse to cold and salt stresses were further analyzesd via qRT-PCR, which provided an experimental support for the conclusions from the RNA-seq datasets. Finally, we have made correction the previous title of “Genome-Wide Identification and Characterization of Key Structural Gene Families Involved in the Biosynthesis of Rice Flavonoid Scaffolds ” to “Genome-Wide Identification and Expression Profiles of 13 Key Structural Gene Families Involved in the Biosynthesis of Rice Flavonoid Scaffolds” for the subject accuracy according to the Reviewer’s comments.

 Point 2: Line 174. Duplication, how to defined? Should add something in the text.

Response 2: Considering the Reviewer’s suggestion, we have added the sentence of “ Gene duplication events including segmental and tandem duplication events are the two principle power to lead to expansion of gene families. Genes in the same subfamily were regarded as co-paralogs. Genes that were co-paralogs and located on duplicated chromosomal blocks within their genomes through polyploidy followed by chromosome rearrangements were defined as segmental duplicates. We defined tandem duplicates as adjacent co-paralogs within the same or neighboring intergenic region on a single chromosome.” in the text.

Point 3: In my downloaded PDF version, there have two Fig.2.

Response 3: New changes in the main text according to the Reviewer’s comments are marked by using the “Track Changes” function in MS Word, which may lead to the results that the reviewer downloaded PDF version still retained the previous unchanged figures such that any changes can
be easily viewed by the editors and reviewers. We have made correction to avoid the same pboblem again in the new manuscript according to the Reviewer’s comments

Point 4: In my downloaded PDF version, Fig.4, Fig.5, and Fig.6 look like strange, 2 legends?

Response 4: Considering the Reviewer’s suggestion, these results have been finally corrected in our new manuscript

Point 5: Line 389, please add the detailed information of each phytohormone, such as concentration, name, etc...

Response 5: We are very sorry for our negligence of the detailed information of each phytohormone treatment, such as concentration, name, etc.These detailed information have been added in the text and figure 7 in the manuscript.

Special thanks to you for your good comments.